# Research Commentary: A Carer’s Roadmap for Research, Practice, and Policy on Suicide, Homicide, and Self-Harm

**DOI:** 10.3390/bs9050048

**Published:** 2019-05-01

**Authors:** Kristin Liabo, Siobhan O’Dwyer

**Affiliations:** 1Peninsula Public Involvement Group (PenPIG), College for Medicine and Health, University of Exeter, St Luke’s Campus, Heavitree Road, Exeter EX1 2LU, UK; PIExeter@exeter.ac.uk; 2College for Medicine and Health, University of Exeter, St Luke’s Campus, Heavitree Road, Exeter EX1 2LU, UK; 3Medical School, College of Medicine and Health, University of Exeter, Exeter EX1 1TX, UK; s.odwyer@exeter.ac.uk

**Keywords:** patient and public involvement, research prioritisation

## Abstract

Academic researchers are increasingly asked to engage with the wider world, both in terms of creating impact from their work, and in telling the world what goes on in university research departments. An aspect of this engagement involves working with patients, carers or members of the public as partners in research. This means working with them to identify important research questions and designing studies to address those questions. This commentary was jointly written by two researchers and people with relevant caring experience for this special issue. It brings to the forefront the concerns of carers who are also involved in research as partners. The aim is to highlight their perspectives to inform future research, policy, and practice.

## 1. Introduction

Academic researchers are increasingly asked to engage with the wider world, both in terms of creating impact from their work, and in telling the world what goes on in university research departments. An aspect of this engagement involves working with patients, carers or members of the public as partners in research. This means working with them to identify important research questions and designing studies to address those questions. People with lived experiences of a medical condition as patients or carers know the topic from a different angle compared to researchers. Together, researchers, patients, carers and providers can design studies that respond to problems as they are experienced in communities, hospitals and primary care settings [1]. 

There is a noticeable absence of such involvement in the existing research on suicide, homicide, and self-harm in family carers. None of the existing studies report having worked with the carers in identifying research questions or developing their study methods, or in the interpretation of their findings. Without the involvement of carers, research in this area may be neglecting key variables, ignoring the most pressing concerns of research participants, and missing opportunities to translate research findings into meaningful policy and practice. While research will always be concerned with appropriate sampling to ensure a study’s validity, involvement of carers in the design and conduct of research brings their perspectives into the analysis of findings and formation of recommendations. This commentary brings to the forefront the concerns of carers who are also involved in research as partners. The aim is to highlight their perspectives to inform future research, policy, and practice. The researchers met individually with carer co-authors to maintain their full anonymity, and they were involved in writing this commentary, based on our conversations.

## 2. Writing This Commentary

The Peninsula Public Involvement Group [PenPIG] was set up by the Collaboration for Leadership in Applied Health Research and Care South West Peninsula (PenCLAHRC). This is one of thirteen applied health research programmes in the UK, funded by the National Institute for Health Research. Because of the applied aspect of this programme, involvement of patients and carers in the development of research is central to every project. PenPIG was established to enable ongoing collaboration with patients who have a special interest in working with researchers, and who have gained experience and knowledge of research through involvement in many different studies. PenPIG is not a representative group of carers, but a group of people with direct experience of caring for adult relatives, children, and friends. 

One of our personal stories concerns an elderly husband caring for his wife after she was diagnosed with a severe and deteriorating brain condition. He struggled with thoughts that a lethal injection would spare his wife’s suffering. As a religious man, he felt guilty for having these thoughts. He also felt that he was failing his wife when he was despairing over her mood and personality changes. To cope, he started self-harming, but he kept this a secret and reassured his children that everything was fine. One day, the vicar came to visit him and found him bleeding from self-inflicted injuries. He was hospitalised and subsequently provided with the support he needed, including respite stays with his grandchildren in another part of the country. This is just one experience of many in our group.

For the purposes of this commentary, PenPIG members were invited to share their experiences of caring and offer recommendations for the future of research on suicide, homicide, and self-harm in family carers. Some members met face-to-face with the researchers, while others provided feedback via email. Members were provided with summaries of existing research and questions to prompt discussion. In reflecting on the existing research and making recommendations for the future, PenPIG members drew on their own experience as well as the stories and experiences of their extended networks of friends, family and acquaintances with whom they have shared common caring experiences. 

The recommendations below were co-written by researchers and carers. As much as possible, the recommendations are presented in carers’ own words and have been edited by the researchers only for clarity or brevity. This roadmap is not representative of all carers, but we hope that by presenting these recommendations from a carer’s perspective, we will encourage researchers in the field to start working directly with carers to develop their research ideas and programmes. 

## 3. Recommendations


*(1) Prevention is better than cure.*


The existing research clearly shows that many carers experience thoughts of suicide and homicide, and that some carers engage in self-harm and attempted suicide and homicide. We believe that there is an urgent need to develop evidence-based prevention strategies to support carers *before* they contemplate suicide, homicide or self-harm. This includes a need for research on the connections between pre-existing mental health problems in carers, before they started caring, and their subsequent risks of homicide, self-harm and suicide. 


*(2) It takes a village.*


All the existing research has focused on individual carers, but we know that true care is rarely done by a single person. We need to know more about the impact of suicide, homicide and self-harm on immediate and extended family members of carers, friends and neighbours. Some of us have witnessed first-hand the ripple effect in local communities and personal networks when someone commits suicide or engages in self-harm. We expect these would be even more powerful if the person was a carer (such as the story above).


*(3) It takes two to tango.*


Carers and the people they care for come as a set, but none of the existing research has considered the perspective of the person receiving care. Research must also consider the people cared for; their own thoughts of suicide, homicide and self-harm; and how this links in with the carer’s thoughts and experiences. Carers’ wellbeing also links to how services are provided. By this, we do not only mean support services to carers, but also how services are provided to the person they care for. For example, carers regularly have to spend much of their time on getting a referral or obtaining help or advice about the person they care for. Getting a referral to a specialist, a physiotherapist, mental health services, a doctor’s appointment, or prescription for those you care for can be a time-consuming ordeal that heightens stress and anxiety in carers and is a contributing factor to negative feelings towards the cared for person.


*(4) Support, support, support.*


Although some people might find the existing research surprising, the high rates of suicidal thinking identified in these studies come as no surprise to us. We firmly believe that the single most effective thing for preventing suicide, homicide and self-harm is more formal support. In the UK, for example, there is not enough support available for family carers, and there is scarce information on the support that is provided. Many carers do not know where to turn or go to for help, and their situation is exacerbated by stress and social isolation from the caring.


*(5) It’s a small world after all.*


Writing from the UK, we are interested in research articles reporting on results from carers in Uganda, Malawi and South Africa, Taiwan and Korea [2,3,4,5]. We would like to know whether there are similar or different challenges facing carers across different cultures. Are there differences in cultural views on suicide, homicide and self-harm? How do people in different cultures draw upon support when they feel stressed or have bad thoughts towards the people they care for? We would welcome more comparative research across low, middle and high income countries, and across continents, perhaps drawing on existing published studies from a whole range of countries and cultures. Perhaps there are things we could learn from each other, across continents?


*(6) Tell us a story.*


We also welcome the small (but hopefully growing) qualitative body of literature that brings to life carers’ personal experiences of suicide, homicide and self-harm [3,6]. Reading stories from other carers can be helpful as it creates a sense of recognition and feeling of being with others, even if they are not in the room. Stories and quotes reported by qualitative studies should inform future research, and could be used to initiate discussions with carers on the triggers that might lead a carer to self-harm; commit suicide; or harm, abuse or yell at the person they care for. This might help with the sensitivity around talking to people about this in a research setting, and also help involve carers in shaping the directions of this research—as we have done with this commentary. 


*(7) Keep our stories safe.*


This is an extremely difficult subject area to research, and one within which carers may have an issue with openness and honesty. The fear of acknowledgment of their own thoughts is problematic, made worse by the fear of judgement and prosecution. Discussion about harm to themselves may be less hard to discuss than their thoughts of harming another. One way around the sensitivity could be to start research with those who are no longer exposed to fear of prosecution or judgement, perhaps people who were carers in the past. Carers are only likely to answer questions honestly if they are in a trusted setting, for example, with a nurse or stranger, and reassured that anything they disclose is confidential, and that they will not get prosecuted, but will get help if anything they disclose warrants immediate concern. Sensitive, non-judgmental questions asked by a person who has awareness and understanding of caring and its difficulties is paramount. Information needs to be provided on why the questions are being asked, and what will happen to the information given by the carer. Carers need to be provided with the option of being interviewed individually or in a group, and with information about services that can help. We appreciate that there are ethical challenges to this work, and encourage researchers to work with carers to address these, including how to break confidentiality if stories of harm are disclosed.

## 4. Conclusions

As carers, we value research on suicide, homicide and self-harm, but our decision to remain anonymous in this paper is a stark reflection of the fear, stigma and misinformation that still exists around both caring and issues of suicide, homicide and self-harm. It is also clear that much remains to be understood about these complex phenomena and how to best address them in policy and practice. The recommendations above come from carers who are familiar with the existing research and have a lived experience of suicidal ideation and self-harm. It is their road map for the future.

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
