# Peer review of "Research Commentary: A Carer’s Roadmap for Research, Practice, and Policy on Suicide, Homicide, and Self-Harm"

_behavsci, 2019, doi:10.3390/bs9050048_

Reviewer 1 Report

Thank you for the important contribution to the research field. The paper is novel and interesting.

However:

1) The ethical considerations should be incorporated in this manuscript. Did the participants agree that their experiences was recognizable or is it anonymized? 

To my knowledge this form of manuscript with findings about the user perspective is not yet established with guidelines or checklists, and therefore I think that the ethical perspective should be stressed.

2) It is difficult for the reader to know whether the results are valid because the methodology is not described thoroughly, e.g. how many members participated in the group? I think that as a peer planning studies I need know to what extent the presented perspectives are representative. 

Author Response

Dear reviewer,

Thank you for your review of our research commentary. 

1) In regards to your comment on the ethical aspects of this commentary, we want to emphasize that this is not a research report. The recommendations were developed jointly between researchers and people with caring experience, through conversations. Our co-authors have chosen to be anonymous due to the stigma attached to this topic. Please note that they were not research participants and we do not claim to report on a piece of research. To emphasize this we have deleted words that refer to their 'lived experience', since we believe this might have caused the conflation with qualitative research.

As you point out, there are ethical implications of discussing sensitive topics, whether this is with people who has experiential knowledge of the topic, professionals or researchers. Thank you for pointing this out. We have added a sentence to explain how we worked around this with the group.

2) We hope that our clarification in regards to this being a co-authored piece rather than a piece of research helps with the generalisability comment. While this commentary is not representative of all carers, it does present new and previously unheard perspectives on research. Usually, research agendas are set by researchers, policy makers, funding organisations and health professionals. This commentary is an attempt to address this imbalance and strengthen the voice of carers in the debate.

Best wishes,

Kristin Liabo on behalf of the authors

Reviewer 2 Report

Thank you for an opportunity to review this manuscript. It's a valuable and innovative contribution to the literature, an example of constructive collaboration between researchers and consumers, which can lead to richer and more "translatable" studies involving carers. It's a though provoking commentary.

A few points need further clarification. 

Writing this commentary: I understand that the work which resulted in the commentary was a "research study"; could the authors provide information regarding the ethics approval for the "data collection". To ensure the rigour of the report, could there be more information provided on the questions asked of  the PenPIG members and the method of qualitative analysis of the collected responses, as well as who conducted the analysis? How many carers took part in the interviews/providede their feedback by email? Also, given the natural diversity of "carers", could some information be provided regarding the type of medical conditions of patients and carers' relationships to them (spouses, children, friends, others)? Also, what kind of "existing research" examples were provided to the participants and what were the criteria of using particular examples? 

I suggest avoiding using the term "completed suicide" because of its possible negative connotations, widely described in the suicidology literature. Also, the everyday term "a physio" could be replaced with "a physiotherapist" in the manuscript. 

Please, clarify the meaning of "many people find existing research shocking..."

"Support, support, support": could you clarify the meaning of "support" in this section? Does it refer to informal and/or formal support?       

If possible, also comment earlier in the text on the "lived experience of suicidal ideation and self-harm" in some of the carers included in the PenPIG group.

I will also appreciate a comment or a reflection regarding "generalisibility" of the presented "roadmap".            

Author Response

Dear Reviewer,

In regards to your first point, this commentary was not based on a research study. The commentary is only just a commentary, with the aim of bringing carers' views into the research debate. We appreciate that this is unusual, and that it is not representative of all carers. However, we argue that because carers' own voice and view on research priorities are usually left out of the discussion, there is merit in publishing their perspectives. We have made some small changes to the wording which we hope clarify that this is not based on a study.

We have deleted 'completed' suicide, and changed physio to the full word. We originally wrote these sentences like this to honour the input from our lay co-authors (carers) but appreciate that this might not be appropriate for an academic journal.

Thank you for highlighting the strange wording of research being 'shocking', we have re-phrased this to fit an academic journal. Again, this reflected the words of the carers involved.

We have re-phrased the section on support to emphasize that this is about formal support.

We have added a sentence about generalisability before presenting the roadmap, which reflects what we have said in this response.

Thank you again for your helpful comments.

Best wishes,

Kristin Liabo on behalf of all authors
